

# Relationship between vertical facial morphology and dental arch measurements in class II malocclusion: a retrospective study

Irmak Ocak[1], Nurver Karsli[2], Ayse Tuba Altug[3] and Muge Aksu[4]

[1] Department of Orthodontics, Faculty of Dentistry, Ankara Medipol University, Ankara, Turkey
[2] Department of Orthodontics, Faculty of Dentistry, Karadeniz Technical University, Trabzon, Turkey
[3] Department of Orthodontics, Faculty of Dentistry, Ankara University, Ankara, Turkey
[4] Unaffiliated, Ankara, Turkey

## ABSTRACT

**Background:** To evaluate the relationship between dental arch measurements and the vertical facial pattern determined in skeletal Class II untreated patients.

**Methods:** Lateral cephalograms and plaster models were obtained from 124 untreated female adults (average age: 17.6 ± 3.8 years). Class I (CI), Class II Division 1 (CII/1) and Class II Division 2 (CII/2) malocclusions were divided into three subgroups according to their vertical morphology as hypodivergent, normodivergent and hyperdivergent. The multivariate variance analysis (MANOVA) method was used in the comparison of measurement values according to vertical and sagittal morphology. The relationship between both A point-Nasion-B point (ANB) and Frankfurt-mandibular plane (FMA) angles and dental arch measurements was examined by Pearson correlation analysis. The significance level was received as $p < 0.05$.

**Results:** While vertical morphology has a statistically significant effect on mandibular arch length, sagittal morphology affects maxillary arch depth. The parameters influenced by both morphologies are maxillary and mandibular arch length, as well as maxillary intermolar width. The mandibular arch length was significantly shorter in hyperdivergent-CII-2 malocclusion (50.5 ± 7.4 mm). Larger values were obtained in both mandibular arch length and maxillary arch depth measurements in CII-1 malocclusion compared to CII-2 malocclusion. The maxillary intermolar width was significantly shorter in hypodivergent-CII-1 malocclusion (46.8 ± 3.4 mm), while it was higher in hypodivergent-CI malocclusion (51.1 ± 3.4 mm). The maxillary arch length was the lowest in hyperdivergent-CI malocclusion (63.1 ± 13.3 mm) and the highest in hypodivergent-CI malocclusion (72.8 ± 7.6 mm). Additionally, a positive but weak correlation was found between ANB and FMA angles.

**Conclusion:** Dental arch measurements have been found to be affected by both vertical facial morphology and skeletal sagittal relationship. A positive correlation was found between ANB and FMA angles.

Corresponding author
Irmak Ocak,
irmakpartal@hotmail.com

# INTRODUCTION

Malocclusion is a condition that needs to be considered three-dimensionally and can be resolved by evaluating many parameters at the same time. Class II (CII) malocclusion is a type of malocclusion accompanied by varying degrees of maxillary protrusion and/or mandibular retrusion and is divided into two subgroups division 1 (CII-1) and division 2 (CII-2) according to incisor inclinations. In addition to the sagittal component, vertical and transversal morphology are very important for the successful treatment of CII malocclusion. Therefore, malocclusion should be assessed with all its dimensions.

Although the literature provides information about the effects of malocclusions on arch structure, the findings of both mandibular and maxillary measurements display variation. Some studies have shown that the maxillary posterior arch width is smaller in CII-1 malocclusion than in Class I (CI) or ideal occlusion (*Sayin & Turkkahraman, 2004*; *Uysal et al., 2005*; *Lux et al., 2003*; *Al-Khateeb & Abu Alhaija, 2006*). *Huth et al. (2007)* found that arch width dimensions in individuals with CII-2 malocclusion were intermediate between those with normal occlusion and CII-1 malocclusion. Another study reported normal dental arch form in CII-2 malocclusion, and only the mandibular intercanine width was reduced (*Walkow & Peck, 2002*). *Slaj et al. (2010)* demonstrated that individuals with CI and CII malocclusion had similar maxillary dental arch dimensions, while only CII individuals had a mandibular transverse deficiency. However, no division classification was made in the Class II malocclusion group in this study.

*Isaacson et al. (1971)* stated that passive stretch tension increases with the elongation of the muscles in a backward-rotated growth pattern, which decreases the maxillary intermolar width. Another study reported that as the mandibular plane-SN (Sella-Nasion plane) angle increases, the mandibular intermolar width also decreases, in addition to the maxilla, thus causing a decrease in maxillary and mandibular arch lengths and that if the decrease in the arc length is not compensated, it will result in crowding, clinically (*Nasby et al., 1972*). *Forster, Sunga & Chung (2008)* and *Khera et al. (2012)* stated that dental arch width was associated with gender and vertical facial morphology, reporting greater vertical dimension with decreasing dental arch width. *Grippaudo et al. (2013)* reported that the vertical facial pattern was only associated with the upper dental arch form in CII malocclusion.

It is very important to know the distinguishing characteristics in terms of planning an accurate treatment and creating a successful retention prescription. When examining CII malocclusion, it is recommended to evaluate not only the craniofacial morphology but also the dentoalveolar features. The aim of this study is to examine the dental arch structure in individuals with CII malocclusion by taking into account vertical facial morphology. Our first null hypothesis was that the difference in vertical facial pattern in CII malocclusion would not affect dental arch parameters. Our second null hypothesis is that there is no correlation between dental arch parameters and sagittal relationship and vertical facial morphology. The predetermined level of significance was set at $p < 0.05$,

suggesting that any *p* value below this threshold would lead to the rejection of the null hypothesis.

## MATERIALS AND METHOD

### Subject selection

In our study, which has a retrospective design, the records of individuals who applied to two Hacettepe University and Ankara University between 2010–2019 were examined. Maxillary and mandibular plaster models and cephalometric radiographs of a total of 124 Anatolian Turkish female individuals (mean age, 17.6 ± 3.8 years) were included. The growth and development stage was determined by the same investigator (I.O.) in cephalometric radiographs using the cervical vertebrae maturation method, as described by *Baccetti, Franchi & McNamara (2005)*.

Inclusion criteria were set as follows: (1) have completed growth and development, (2) presence of all permanent teeth excluding third molars, (3) not having impacted teeth, (4) not having supernumerary teeth, (5) not having any syndromes, and (6) not having undergone any previous orthodontic treatment. Patients who met the inclusion criteria were then classified according to their malocclusions by evaluating their sagittal relationship. A bilateral Class 1 molar relationship with an ANB (A point-Nasion-B point) angle between 0–4 degrees was considered CI malocclusion, while a bilateral Class 2 molar relationship with an ANB angle above four degrees was considered CII malocclusion. Individuals with CII malocclusion were divided into divisions according to incisor relationships. Then, vertical morphology was evaluated according to the FMA (Frankfurt-mandibular plane) angle, and subjects were divided into three different groups: hypodivergent (<22 degrees), normodivergent (between 22–28 degrees) and hyperdivergent (>28 degrees). FMA angle has been preferred as it is one of the most reliable indicators in assessing vertical growth pattern and is not affected by cranial base discrepancies (*Ahmed, Shaikh & Fida, 2016*).

Exclusion criteria were as follows: (1) cervical maturation stage of five and below according to growth and development criteria, (2) crowding greater than 9 mm, (3) presence of unilateral or bilateral crossbite, (4) history of trauma, (5) presence of tooth wear that changes the size and form of the teeth, (6) presence of dental prosthesis that may prevent measurements, (7) missing plaster model and/or radiographic records.

Our study was approved by the Ethics Committee of Hacettepe University (Approval number: GO-19/646). The authors declare that there is no conflict of interest.

### Dental arch measurements

Measurements were made manually on plaster models using the digital vernier caliper (precision 0.01 mm). Calibration was checked before each measurement. All measurements were performed by the first author (I.O.), and the measurements were repeated for reliability after 15 days by randomly selecting 24 plaster models. The dental arch measurements used in the study are as follows for both maxilla and mandible (Fig. 1):

 

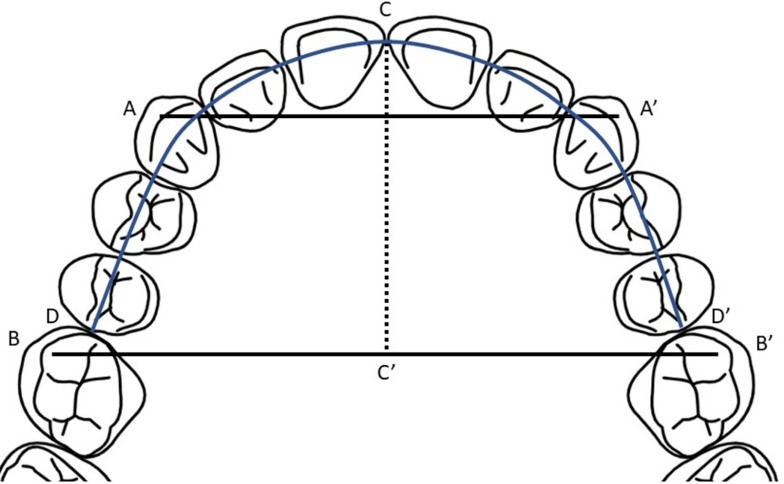

**Figure 1** **Dental arch measurements on plaster model.**

- Intercanine width (A-A′): Linear distance between the cusp tips of right and left permanent canines.
- Intermolar width (B-B′): Linear distance between the mesiobuccal cusp tips of right and left permanent first molars.
- Arch length (CD+CD′): Sum of the linear distances from the incisal point to mesial surfaces of the permanent first molars.
- Arch depth (C-C′): Linear perpendicular distance from the incisal point to the intermolar width.

### Statistical analysis

All data were analyzed by using the SPSS 23.0 statistical software program (SPSS Inc, Chicago, IL, USA). Conformity to normal distribution was assessed with the Shapiro–Wilk test. Intra-observer reliability was evaluated with the intra-class correlation coefficient. Multivariate analysis of variance (MANOVA) was used to compare measurements according to vertical and sagittal morphology. The Bonferroni test was used for multiple comparisons. Quantitative data were presented as mean and standard deviation. The relationship between the ANB and FMA angles and dental arch measurements was analyzed using Pearson's correlation analysis. The *post-hoc* power analysis was performed using G*Power software (version 3.1.9.7; Heinrich-Heine University, Düsseldorf, Germany) and the significance level was set at $p < 0.05$.

### RESULTS

The intra-class correlation coefficient was calculated to be between 0.975–0.995 for maxillary measurements and 0.965–0.996 for mandibular measurements. Repeated measurements demonstrated a high degree of consistency. Descriptive statistics are shown in Table 1. *Post hoc* power analysis revealed an effect size (f) of 0.51 and statistical power

**Table 1 Descriptive statistics of the participants.**

|  | Group | Hypodivergent | Normodivergent | Hyperdivergent |
|---|---|---|---|---|
| N (%) | CII-1 | 15 (38.5%) | 18 (41.9%) | 24 (57.1%) |
|  | CII-2 | 10 (25.6%) | 8 (18.6%) | 4 (9.5%) |
|  | CI | 14 (35.9%) | 17 (39.5%) | 14 (33.4%) |
|  | Total | 39 (100%) | 43 (100%) | 42 (100%) |
| Age (yr) | CII-1 | 16.83 ± 2.29 | 18.04 ± 4.37 | 17.65 ± 3.52 |
|  | CII-2 | 16.43 ± 2.28 | 17.54 ± 2.42 | 16.15 ± 1.46 |
|  | CI | 18.82 ± 7.94 | 17.18 ± 2.07 | 18.01 ± 2.04 |
|  | Total | 17.44 ± 5.09 | 17.61 ± 3.23 | 17.63 ± 2.95 |
| ANB (°) | CII-1 | 5.5 ± 1.3 | 6.6 ± 1.6 | 7.5 ± 1.6 |
|  | CII-2 | 6.5 ± 1.3 | 6 ± 1.3 | 7.5 ± 1.4 |
|  | CI | 2.6 ± 1.5 | 2.8 ± 1.1 | 3.5 ± 0.5 |
|  | Total | 4.7 ± 2.1 | 5 ± 2.3 | 6.2 ± 2.3 |
| FMA (°) | CII-1 | 18.3 ± 2.4 | 26.1 ± 1.4 | 31.6 ± 2.8 |
|  | CII-2 | 17.4 ± 4.1 | 25.2 ± 1.5 | 33.6 ± 4.7 |
|  | CI | 18.5 ± 3.0 | 24.5 ± 1.3 | 31.1 ± 2.3 |
|  | Total | 18.2 ± 3.1 | 25.3 ± 1.5 | 31.7 ± 2.9 |
| Overjet (mm) | CII-1 | 5.9 ± 3 | 4.9 ± 2.1 | 5.4 ± 3.1 |
|  | CII-2 | 3.5 ± 1.1 | 4.2 ± 1.6 | 4.1 ± 2.9 |
|  | CI | 3 ± 1.5 | 2.7 ± 1.3 | 3.2 ± 1.2 |
|  | Total | 4.2 ± 2.5 | 3.9 ± 2 | 4.5 ± 2.7 |
| Overbite (mm) | CII-1 | 3.8 ± 2.3 | 3 ± 2.2 | 1.0 ± 2.5 |
|  | CII-2 | 9.1 ± 2.7 | 5.5 ± 1.8 | 5.4 ± 1.7 |
|  | CI | 2.3 ± 3 | 1.3 ± 2.8 | 0.3 ± 2.5 |
|  | Total | 4.6 ± 3.8 | 2.8 ± 2.6 | 1.2 ± 2.8 |

**Notes:**
Values are presented as mean ± standard deviation or number (%).
ANB, A point-nasion-B point angle; FMA, Frankfurt-mandibular plane angle; Overjet, the horizontal distance between the incisal tips of the maxillary central incisor and the buccal surface of the mandibular central incisor; Overbite, vertical distance between the incisal tips of the maxillary and mandibular central incisors; CII-1, Class II division 1 malocclusion; CII-2, Class II division 2 malocclusion; CI, Class I malocclusion.

was 0.971 as calculated with α error = 0.5, showing that the sample size was adequate in this study.

The MANOVA results demonstrated that the vertical morphology had a statistically significant effect only on the mandible arch length (Table 2, $p = 0.012$). There was a statistically significant difference in mandible arch length between normodivergent and hyperdivergent groups according to vertical morphology. The mandibular arch length was significantly shorter in the hyperdivergent group (Table 3, $p = 0.01$).

Sagittal morphology groups showed statistically significant effects on maxillary arch depth and mandibular arch length (Table 2, $p = 0.001$ and $p = 0.018$, respectively). There was a statistically significant difference between CII-1 and CII-2 malocclusion in both measurements. The CII-1 malocclusion group exhibited greater values for both

**Table 2  Results of MANOVA.**

| | P value | | |
| | Vertical morphology | Sagittal morphology | Vertical * Sagittal morphology |
|---|---|---|---|
| Maxilla | | | |
| Intercanine width (mm) | 0.848 | 0.471 | 0.050 |
| Intermolar width (mm) | 0.282 | 0.186 | 0.015* |
| Arch length (mm) | 0.529 | 0.261 | 0.006** |
| Arch depth (mm) | 0.898 | 0.001** | 0.074 |
| Mandible | | | |
| Intercanine width (mm) | 0.774 | 0.537 | 0.105 |
| Intermolar width (mm) | 0.056 | 0.522 | 0.071 |
| Arch length (mm) | 0.012* | 0.018* | 0.004** |
| Arch depth (mm) | 0.543 | 0.076 | 0.078 |

Notes:
* $P < 0.05$.
** $P < 0.01$

**Table 3  Mandibular dental arch measurements.**

| | | Hypodivergent | Normodivergent | Hyperdivergent | Total |
|---|---|---|---|---|---|
| Mandible | | | | | |
| Intercanine width (mm) | CII-1 | 25.4 ± 2.2 | 25.7 ± 1.9 | 26.6 ± 1.5 | 26.0 ± 1.9 |
| | CII-2 | 24.9 ± 3.3 | 25.1 ± 1.8 | 26.2 ± 3.3 | 25.2 ± 2.7 |
| | CI | 26.3 ± 2.6 | 25.4 ± 2.1 | 24.6 ± 2.9 | 25.4 ± 2.6 |
| | Total | 25.6 ± 2.7 | 25.5 ± 1.9 | 25.9 ± 2.3 | 25.7 ± 2.3 |
| Intermolar width (mm) | CII-1 | 43.0 ± 3.5 | 43.6 ± 2.2 | 43.8 ± 2.4 | 43.5 ± 2.6 |
| | CII-2 | 41.2 ± 3.4 | 45.1 ± 2.7 | 42.1 ± 3.8 | 42.8 ± 3.6 |
| | CI | 44.5 ± 3.4 | 44.6 ± 3.8 | 42.3 ± 4.1 | 43.8 ± 3.8 |
| | Total | 43.1 ± 3.6 | 44.3 ± 3.0 | 43.1 ± 3.2 | 43.5 ± 3.3 |
| Arch length (mm) | CII-1 | 59.3 ± 8.1[AB] | 61.9 ± 5.6[AB] | 63.3 ± 5.2[B] | 61.8 ± 6.3[b] |
| | CII-2 | 57.8 ± 8.8[AB] | 61.0 ± 3.8[AB] | 50.5 ± 7.4[A] | 57.6 ± 7.8[a] |
| | CI | 64.0 ± 6.6[B] | 63.1 ± 3.4[B] | 56.3 ± 11.2 | 61.3 ± 8.1[b] |
| | Total | 60.6 ± 8.0[ab] | 62.2 ± 4.5[b] | 59.8 ± 8.9[a] | 60.9 ± 7.4 |
| Arch depth (mm) | CII-1 | 22.7 ± 2.7 | 23.6 ± 3.3 | 23.6 ± 1.8 | 23.3 ± 2.5 |
| | CII-2 | 21.4 ± 3.5 | 22.1 ± 2.8 | 21.7 ± 2.1 | 21.7 ± 2.9 |
| | CI | 23.8 ± 2.3 | 22.8 ± 2.1 | 20.9 ± 3.4 | 22.5 ± 2.8 |
| | Total | 22.8 ± 2.9 | 23.0 ± 2.8 | 22.5 ± 2.7 | 22.8 ± 2.8 |

Notes:
Values are presented as mean ± standard deviation.
a-b: The same letters indicate that there were no statistically significant difference between main groups, A-B: The same letters indicate that there were no statistically significant difference between interaction groups.
CII-1, Class II division 1 malocclusion; CII-2, Class II division 2 malocclusion; CI, Class I malocclusion.

measurements (Table 3, $p = 0.002$ for mandibular measurement and Table 4, $p = 0.019$ for maxillary measurement). In addition, a statistically significant difference was found in mandibular arch length between CI and CII-2 malocclusion groups (Table 3, $p = 0.041$).

**Table 4 Maxillary dental arch measurements.**

| | Group | Hypodivergent | Normodivergent | Hyperdivergent | Total |
|---|---|---|---|---|---|
| Maxilla | | | | | |
| Intercanine width (mm) | CII-1 | 31.4 ± 3.0 | 32.9 ± 2.1 | 32.8 ± 2.0 | 32.5 ± 2.4 |
| | CII-2 | 32.9 ± 4.2 | 32.1 ± 2.1 | 34.2 ± 1.8 | 32.8 ± 3.2 |
| | CI | 33.8 ± 3.1 | 32.0 ± 2.9 | 30.5 ± 3.3 | 32.1 ± 3.3 |
| | Total | 32.6 ± 3.4 | 32.4 ± 2.4 | 32.2 ± 2.8 | 32.4 ± 2.9 |
| Intermolar width (mm) | CII-1 | 46.8 ± 3.4$^B$ | 49.1 ± 2.7$^{AB}$ | 48.4 ± 2.6$^{AB}$ | 48.2 ± 2.9 |
| | CII-2 | 47.3 ± 3.8$^{AB}$ | 49.8 ± 2.1$^{AB}$ | 48.9 ± 4.3$^{AB}$ | 48.5 ± 3.5 |
| | CI | 51.1 ± 3.4$^A$ | 49.4 ± 3.2$^{AB}$ | 47.5 ± 4.7$^{AB}$ | 49.3 ± 4.0 |
| | Total | 48.5 ± 4.0 | 49.3 ± 2.8 | 48.1 ± 3.5 | 48.7 ± 3.4 |
| Arch length (mm) | CII-1 | 67.3 ± 9.3$^{AB}$ | 71.4 ± 6.3$^{AB}$ | 72.7 ± 4.2$^B$ | 70.9 ± 6.8 |
| | CII-2 | 66.7 ± 10.3$^{AB}$ | 67.9 ± 2.9$^{AB}$ | 67.2 ± 11.6$^{AB}$ | 67.2 ± 8.2 |
| | CI | 72.8 ± 7.6$^B$ | 70.5 ± 3.4$^{AB}$ | 63.1 ± 13.3$^A$ | 68.9 ± 9.5 |
| | Total | 69.1 ± 9.2 | 70.4 ± 4.9 | 69.0 ± 9.8 | 69.5 ± 8.2 |
| Arch depth (mm) | CII-1 | 27.2 ± 4.0 | 28.2 ± 3.0 | 28.3 ± 1.9 | 28.0 ± 2.9$^a$ |
| | CII-2 | 24.5 ± 5.0 | 24.4 ± 2.2 | 25.7 ± 4.3 | 24.7 ± 3.9$^b$ |
| | CI | 28.0 ± 2.7 | 26.8 ± 1.6 | 24.6 ± 4.4 | 26.5 ± 3.3$^{ab}$ |
| | Total | 26.8 ± 4.1 | 26.9 ± 2.7 | 26.8 ± 3.6 | 26.8 ± 3.4 |

Notes:
Values are presented as mean ± standard deviation.
a-b: The same letters indicate that there were no statistically significant difference between main groups, A-B: The same letters indicate that there were no statistically significant difference between interaction groups.
CII-1, Class II division 1 malocclusion; CII-2, Class II division 2 malocclusion; CI, Class I malocclusion.

A statistically significant interaction was observed between vertical and sagittal morphology among maxillary intermolar width, maxillary arch length, and mandibular arch length measurements (Table 2, $p = 0.015$, $p = 0.006$ and $p = 0.004$, respectively).

A statistically significant difference was found between hypodivergent-CII-1 malocclusion and hypodivergent-CI malocclusion for maxillary intermolar width. Hypodivergent-CII-1 malocclusion was found to have the lowest maxillary intermolar width (Table 4, $p = 0.017$).

Hyperdivergent-CI malocclusion for maxillary arch length showed a statistically significant difference compared to hypodivergent-CI malocclusion and hyperdivergent-CII-1 malocclusion (Table 4, $p = 0.033$ and $p = 0.011$, respectively). Hypodivergent-CI malocclusion displayed the highest value, while hyperdivergent-CI malocclusion displayed the lowest mandibular arch length.

Mandibular arch length measurements showed that the hyperdivergent-CII-2 malocclusion was statistically significantly different from the hypodivergent-CI, normodivergent-CI, and hyperdivergent-CII-1 malocclusions (Table 3, $p = 0.020$, $p = 0.032$, and $p = 0.021$, respectively). The hyperdivergent CII-2 malocclusion was the lowest compared to other subgroups.

The Pearson correlation analysis revealed no statistically significant relationship between any variable and ANB and FMA angle measurements. Only a weak positive correlation was determined between ANB and FMA angles ($r = 0.241$, $p = 0.007$).

## DISCUSSION

In the present study, we investigated the dental arch parameters of individuals with untreated CI and CII malocclusions with different vertical morphologies. We thereby had the opportunity to evaluate the descriptive features of CII malocclusion from multiple perspectives. Our first null hypothesis was rejected for mandibular arch length and all maxillary measurements. When we evaluated the correlation between measurements and facial morphologies, our second null hypothesis was accepted.

In previous studies examining arch parameters by gender, it was shown that dental arch widths were statistically significantly greater in males than female patients (*Forster, Sunga & Chung, 2008*; *Khera et al., 2012*; *Eroz, Ceylan & Aydemir, 2000*; *Bishara et al., 1997*; *Oliva et al., 2018*). In other studies, while no difference was observed in angular measurements between the genders, it was reported that there was a significant difference in linear measurements, especially in high-angle individuals (*Chung & Wong, 2002*; *Chung & Mongiovi, 2003*). Therefore, only female patients were included in the current study so that the results were not affected.

Comparing the dental arch structure of an individual with well-aligned occlusion without crowding, rotation and diastema *etc.* with an individual with CII malocclusion makes the interpretation of the results very difficult. It is also difficult to find a group of patients with ideal occlusion accompanied by vertical abnormalities. Due to these criteria, CI malocclusion was preferred instead of ideal occlusion as the control group in our study. Similar to our study, comparison with CI malocclusion was preferred in many studies (*Walkow & Peck, 2002*; *Slaj et al., 2010*; *Buschang, Stroud & Alexander, 1994*; *Brezniak et al., 2002*).

During the measurements of plaster models, performing manual measurements with a digital caliper was preferred. Dental plaster models are cost-effective and accepted as the gold standard (*De Luca Canto et al., 2015*). However, although direct measurements on plaster models have been reported to be more accurate and reproducible, minor differences can occur with slight variations in caliper-positioning (*Zilberman, Huggare & Parikakis, 2003*). In our study, measurements were made by only one experienced orthodontist, and some of the measurements were randomly repeated after 15 days, with high consistency obtained in repeated measurements.

While the mandibular arch length was the lowest in the hyperdivergent group, the lowest length was found especially in the hyperdivergent CII-2 malocclusion. In their study, *Khera et al. (2012)* examined individuals with CI malocclusion and reported statistically significantly decreased measurements for intercanine, interpremolar and intermolar width in the hyperdivergent group, while no significant difference was found in arch length. Conversely, the findings of *Nasby et al. (1972)* are consistent with our study. However, sagittal morphology was not considered in their research. In our study, although the highest value was observed in CII-1 malocclusion and the lowest value in CII-2 malocclusion, the highest mandibular arch length was observed in the hypodivergent-CI group when vertical morphology was included in the evaluation. Similarly, *Buschang, Stroud & Alexander (1994)* reported the smallest mandible size in CII-2 malocclusion,

followed by the CI malocclusion with the largest in CII-1 malocclusion. They interpreted this situation as an adaptation of the posteriorly positioned mandible trying to align with the narrow and elongated maxilla. In another study, no significant difference was found between CI and CII malocclusion (*Al-Khateeb & Abu Alhaija, 2006*).

Evaluation of the measurements of the maxilla revealed a statistically significant difference in the maxillary intermolar width, maxillary arch length, and maxillary arch depth. While maxillary arch depth was associated with sagittal morphology, the interaction of sagittal and vertical morphology displayed statistical significance for maxillary intermolar width and maxillary arch length.

The present study observed a statistically significant difference for CII-1 and CII-2 malocclusion in maxillary arch depth. Maxillary arch depth was found to be greater in CII-1 malocclusion than in CII-2 malocclusion. In parallel with our findings, *Buschang, Stroud & Alexander (1994)* reported narrow and long maxillary arches in CII-1 malocclusions and wide and short maxillary arches in CII-2 malocclusions. This finding is thought to be due to the difference in the maxillary arch forms and the difference in the incisor inclinations.

Only a statistically significant difference was found between hypodivergent-CII-1 and hypodivergent-CI malocclusion in maxillary intermolar width. The smallest value was measured in hypodivergent-CII-1 malocclusion, while the largest value was measured in hypodivergent-CI malocclusion. *Khera et al. (2012)* showed that the intermolar width was significantly smaller in hyperdivergent individuals than in hypodivergent individuals, which is not compatible with our findings. In another study, it was reported that vertical morphology was not significantly associated with the maxillary intermolar width (*Forster, Sunga & Chung, 2008*). This report is consistent with our findings. In both studies, findings were reported for female subjects, but only subjects with skeletal and dental CI patterns were included. The masticatory muscles have an effect on craniofacial growth. *Tircoveluri et al. (2013)* reported a decrease in vertical dimension and an increase in transverse growth as the masseter muscle thickness increased, and they observed a positive correlation with maxillary intermolar width. In most studies, the intermolar width in CII-1 malocclusion was reported to be smaller compared to both normal occlusion and CI malocclusion (*Sayin & Turkkahraman, 2004*; *Lux et al., 2003*; *Al-Khateeb & Abu Alhaija, 2006*; *Huth et al., 2007*; *Grippaudo et al., 2013*; *Shu et al., 2013*; *Staley, Stuntz & Peterson, 1985*). It has been stated that the decrease in the intermolar width in CII-1 malocclusion results from the palatal inclination of the posterior teeth to provide interdigitation with the mandibular teeth (*Sayin & Turkkahraman, 2004*; *Shu et al., 2013*). *Staley, Stuntz & Peterson (1985)* attributed this to the palatal inclination of the maxillary posterior teeth and the narrow maxillary alveolar base to compensate the buccal overjet and perioral muscle activities. *Uysal et al. (2005)* reported that there is a wider maxillary intermolar distance in CII-1 malocclusion than normal occlusion and that this is due to the buccal inclination of the molar teeth to compensate for the narrow maxilla. Some studies reported no statistical significance in maxillary intermolar width between malocclusions (*Slaj et al., 2010*; *Hajeer, 2014*). Based on these data, it can be said that evaluating sagittal and vertical morphology together yields different results than evaluating each one individually. Furthermore, the

influence of the transverse component of occlusion on sagittal and vertical morphology should not be disregarded.

Among maxillary arch length measurements, the lowest value was found in hyperdivergent-CI malocclusion, while the highest value was found in hypodivergent-CI malocclusion. Our findings are consistent with those of *Khera et al. (2012)*, in which patients with CI malocclusion were examined. In addition, although sagittal relationship was not mentioned, *Nasby et al. (1972)* reported that the maxillary arch length was smaller in hyperdivergent cases, which is consistent with our findings. Interestingly, the present study showed increased maxillary arch length in hyperdivergent-CII-1 malocclusion. This finding was expectable, although it initially seems to contradict our results. Essentially, it is an expected result that the maxillary arch length is increased in individuals with CII-1 malocclusion due to maxillary incisor proclination. The results of *Al-Khateeb & Abu Alhaija (2006)* also support this finding.

*Isaacson et al. (1971)* described that as the face height increases, the muscles lengthen, and the increase in this muscle elongation leads to an increase in passive stretch tension, which will have a constricting effect on the jaws. In a study examining the effects of vertical morphology on dentition, it was reported that there was a statistically significant negative relationship in the maxillary canine, first premolar and first molar regions in male patients and only in first molar widths in female patients (*Khera et al., 2012*). In another study, it was shown that there was a significant but weak relationship between vertical morphology and dental arch width. However, this relationship was limited to the premolar region in female subjects (*Forster, Sunga & Chung, 2008*). In the current study, while no significant relationship was found between dental arch parameters and sagittal and vertical morphology, only a weak positive correlation was found between ANB and FMA angles. *Plaza et al. (2019)* also found a statistically significant relationship between vertical growth pattern and ANB angle, which is consistent with previous studies indicating that individuals with CII malocclusion exhibit more hyperdivergent growth patterns (*Proffit, Fields & Moray, 1998*; *De La Cruz et al., 1995*).

The present study examines dental arch morphology regarding the influence of skeletal structures. Sample size and selection are very substantial in studies examining dental arch parameters. One of the limitations of our study is that we could not include all subjects in the study regardless of gender, which reduced our sample size. Another limitation of our study is that due to its cross-sectional design, features in adulthood were examined, and changes during the growth and development process could not be observed. Conducting more comprehensive and longitudinal diagnostic studies in the future will be beneficial. By this means, the etiological factors will be revealed more clearly and will contribute to a better understanding of the head and neck development process.

## CONCLUSIONS

According to our study, dental arch parameters are affected by both sagittal and vertical morphology. While the mandibular arch length was measured as the smallest in hyperdivergent-CII-2 malocclusion and the largest in hypodivergent-CI malocclusion, the maxillary arch depth was measured as the largest in CII-1 malocclusion. The maxillary

intermolar width is the narrowest in hypodivergent-CII-1 malocclusion and the widest in hypodivergent-CI malocclusion. Maxillary arch length is the smallest in hyperdivergent-CI malocclusion and the greatest in hypodivergent-CI malocclusion. Dental arch form is a parameter that is affected by many factors and should be carefully evaluated. The clinical significance of our study lies in its contribution to our understanding of the relationship between malocclusion and the transverse components of occlusion. By comprehending the variations in dental arch parameters among different malocclusion types, it becomes possible to enhance treatment planning and decision-making processes. For instance, the identification of specific arch characteristics can guide orthodontic interventions aimed at achieving optimal dental alignment and occlusion. In order to ensure stability during clinical applications, it will be useful to take the individual arch form before the treatment as a guide. At the same time, it should be emphasized that the vertical dimension should be taken into account in CII individuals.

### Funding
The authors received no funding for this work.

### Competing Interests
Muge Aksu works in her private clinic. The other authors declare that they have no competing interests.

### Author Contributions
- Irmak Ocak conceived and designed the experiments, performed the experiments, analyzed the data, prepared figures and/or tables, and approved the final draft.
- Nurver Karsli conceived and designed the experiments, analyzed the data, prepared figures and/or tables, and approved the final draft.
- Ayse Tuba Altug conceived and designed the experiments, authored or reviewed drafts of the article, and approved the final draft.
- Muge Aksu conceived and designed the experiments, authored or reviewed drafts of the article, and approved the final draft.

### Ethics
The following information was supplied relating to ethical approvals (*i.e.*, approving body and any reference numbers):

The study was approved by the Ethical Committee of Hacettepe University (Ankara, Turkey) (GO-19/646).

### Data Availability
The raw data are available in the Supplemental File.

## Supplemental Information

Supplemental information for this article can be found online at http://dx.doi.org/10.7717/peerj.16031#supplemental-information.

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
