# Peer review of "Relationship between vertical facial morphology and dental arch measurements in class II malocclusion: a retrospective study"

_PeerJ, doi:10.7717/peerj.16031_

## Round 0.1 · original submission · Minor Revisions

Dear Dr Ocak,

Thank you for your submission to PeerJ; the article's content is within the scope of PeerJ.

·

Basic reporting

Thank you for providing this article. The article refers to appropriate references and thereby provides context. The article is clearly structured. The data has been made available, and the results were well summarized. The main results are provided in tables.

Regarding basic reporting, I want to make the following comments:
1.) In line 129, a placeholder "(xx)" should be replaced with the abbreviation of the corresponding author.
2.) In lines 162 and 170 the term "ve" can be found and should be replaced by "and".
3.) The acronyms SN angle, ANB and FMA were only spelled out as table captions. I propose to add these terms spelled out in the main text once.
4.) In table 1 and 4 there are typographical errors in the column headings. "Divergant" should be replaced with "divergent".
5.) The phrase "arch width dimensions varied between normal occlusion and CII-1 malocclusion in individuals with CII-2 malocclusion" in line 77 is difficult to understand. The findings of Huth et al. might be rephrased as "Maxillary arch width was largest in the CII-2 group and smallest in the CII-1 group, with the normal occlusion group in between."

Experimental design

I have no comments to make regarding the experimental design.

Validity of the findings

Based on the provided data and description of methods, the results can be evaluated. In this regard, I want to make the following comment:
6.) The patient in line 38 of the data sheet has an FMA of 22.4 and has been classified as hypodivergent. According to the classification in lines 117 and 118 of the manuscript, the patient should be described as a normodivergent.

·

Basic reporting

Sufficient field background. Contemporary literature references - the most recent is from 2019.
Professional article structure.

Experimental design

The research hypothesis is well defined.

Validity of the findings

The statistics are well explained.

Additional comments

I would like to congratulate the authors team for the extremely useful in a practical way research.

Reviewer 3 ·

Basic reporting

Page 7, line 75-76: please improve the syntax to make the meaning more understandable
Page 7, line 92: please choose a synonym for "features"

Experimental design

Page 8, line 114-116: explain why FMA was chosen as the divergence control angle over other angles such as SN/mandibular plane, include literature to support the choice.

Page 8, line 117, “exclusion criteria”: explain more precisely whether patients with mono or bilateral cross-bites were excluded from the study, or whether no distinction was made in this respect. please also state whether single-sided second classes were excluded. specify whether or not asymmetrical patients were included in the sample

Validity of the findings

Page 11, line 213: briefly enter which parameters to make the meaning of the citation clearer
Page 11, line 234-237: stress and explain the result from a clinical point of view, e.g. a posterior mandibular adaptation would explain the reduced maxillary width in second-class hypodivergent subjects compared to first-class hypodivergent subjects who show a larger maxillary diameter. also better explain the characteristics of maxillary width in hyperdivergent subjects in first and second class with a different mandibular adaptation.

Additional comments

In table 1 correct '-divergant', and please improve punctuation from line 3 to 7
In the references section add more recent bibliography regarding arch form and divergence (e.g.: Three-dimensional analysis of dental arch forms in Italian population December 2018 Progress in Orthodontics 19(1) DOI:10.1186/s40510-018-0233-1)

Reviewer 4 ·

Basic reporting

1. In the Results of the Abstract section please provide the significant results and give the values, do not only indicate the differences.
2. Line 103: which were these Universities? Please address them.
3. You need to add some more recent references to your bibliography.

Experimental design

1. At the end of the Introduction section, following the null hypotheses, please report the threshold value for significance (p).
2. Set up just one level of significance and be consistent with it for the results of the study.

Validity of the findings

1. You stated you included only female to not affect results, given the previous published data about the difference in the arch dimensions between males and females. This is a great weakness of the study in my opinion. In order to actually be able to state what you stated, you should have included males and females and avelauted both to draw general results. Instead, including only females, your results and so conclusions are restricted to the female gender, you are not allowed to generalise. Also nationality of the participants should be addressed in the discussion section.
2. Lines 203-207: the following, recent, systematic review stated that the validity of measurements obtained after using a laser scanner from plaster models is similar to direct measurements. Please clarify.
- De Luca Canto, G., Pachêco-Pereira, C., Lagravere, M. O., Flores-Mir, C., & Major, P. W. (2015). Intra-arch dimensional measurement validity of laser-scanned digital dental models compared with the original plaster models: a systematic review. Orthodontics & Craniofacial Research, 18(2), 65–76. doi:10.1111/ocr.12068
3. Which is the clinical significance of your study? Please report them.

Reviewer 5 ·

Basic reporting

The reporting of this study is very clear. No problems have been encountered regarding the use of the English language. The writing up is very sound. The literature review is very good. However, the authors have been asked to cite this very relevant reference:
Hajeer MY. Assessment of dental arches in patients with Class II division 1 and division 2 malocclusions using 3D digital models in a Syrian sample. Eur J Paediatr Dent. 2014 Jun;15(2):151-7. PMID: 25102466.

The structure of the paper is well done. Only one Figure has been inserted, illustrating the measurements made on the study model. The Figure is sharp enough, but the legend should be expanded to give the meanings of the letters used within the Figure.

The sharing of all underlying data is clear, and no problems have been found in this issue.
The reliability of the method employed has been assessed in this study, which an important aspect of any research work using measuring methods.

Experimental design

The aims of this work have been clearly announced, and they are very relevant to the scope of this journal. The research question is well-defined. The introduction with the relevant references justifies the onset of this word. Maybe one relevant study should be included in the Introduction or/and the Discussion section:
Hajeer MY. Assessment of dental arches in patients with Class II division 1 and division 2 malocclusions using 3D digital models in a Syrian sample. Eur J Paediatr Dent. 2014 Jun;15(2):151-7. PMID: 25102466.

The following points are related to the Materials and Methods section:
1- What kind of malocclusion was there in the Class I group?
2- Can a patient with an ANB angle of 1 degree be included in the Class I group? Do you have a reference to support your answer?
3- Could you please provide a reference to support your criterion regarding the maturation of the cervical vertebra?
4- Give the product information regarding the G*power program.
5- What kind of sampling did you use to create these three groups in this retrospective study?

Validity of the findings

The findings are very interesting. Many of them are consistent with previous studies.
Please, give more information about the characteristics of the three groups. In other words, what type of deformity was in the Class I group?

The tables are presented in a very clean and neat manner.
The language used to describe the results is very organized.
The tables are informative and well organized and presented in a logical way.
The statistical analysis is professionally done.

---

## Round 0.2 · accepted · Accept

Dear Authors

In agreement with the reviewers, I believe that your manuscript can be accepted for publication.

Congratulations!

Please note the minor corrections from Reviewer 1.

·

Basic reporting

Thank you for providing the revised manuscript.

Please consider the typo in line 262 "compansate" which should be replaced by "compensate" and a redundant colon at the beginning of the results section in the abstract.

Experimental design

Regarding the experimental design, I have no comments to make.

Validity of the findings

Thank you for explaining and solving the issue regarding the FMA and corresponding classification of one patient record in the raw data.

Additional comments

Except for the two minor comments in the basic reporting section, I have no further remarks to make.

Reviewer 3 ·

Basic reporting

I have no additional comments on the revised version.

Experimental design

I have no additional comments on the revised version.

Validity of the findings

I have no additional comments on the revised version.

Additional comments

I have no additional comments on the revised version.

Reviewer 4 ·

Basic reporting

No any other comments

Experimental design

No any other comments

Validity of the findings

No any other comments

Additional comments

No any other comments

Reviewer 5 ·

Basic reporting

In this revised version, everything looks clean and neat.
My raised points in the previous review of this paper have been addressed.

Experimental design

The experimental design is good.
All my raised points in the previous review have been addressed.

Validity of the findings

The findings are valid.
No problems have been encountered regarding the presented tables.

Additional comments

Good paper.